# The Effect of By-Pass Linseed Oil Supplementation on the Maternal Antioxidant System during the Embryo-Maternal Recognition Period in Ewes

**DOI:** 10.3390/ani13162565

**Published:** 2023-08-09

**Authors:** Ignacio Contreras-Solís, Valeria Pasciu, Cristian Porcu, Francesca D. Sotgiu, Neda Todorova, Elena Baralla, Laura Mara, Marilia Gallus, Andrea Cabiddu, Maria Dattena, José Alfonso Abecia, Fiammetta Berlinguer

**Affiliations:** 1Department of Veterinary Medicine, University of Sassari, 07100 Sassari, Italy; cporcu@uniss.it (C.P.); fdsotgiu@uniss.it (F.D.S.); neda_todorova_87@abv.bg (N.T.); ebaralla@uniss.it (E.B.); berling@uniss.it (F.B.); 2Department of Animal Science, AGRIS Sardegna, Loc. Bonassai, 07100 Sassari, Italy; lmara@agrisricerca.it (L.M.); mgallus@agrisricerca.it (M.G.); acabiddu@agrisricerca.it (A.C.); mdattena@agrisricerca.it (M.D.); 3Veterinary Faculty, Zaragoza University, 50013 Zaragoza, Spain; alf@unizar.es

**Keywords:** linseed oil, antioxidant system, reproductive tissues, polyunsaturated fatty acids, mature-dry ewes

## Abstract

**Simple Summary:**

The beneficial effect of polyunsaturated fatty acids (PUFA) in different body systems and their function in mammals is well known. The use of linseed oil (LO, rich in alpha-linolenic acid/ALA) as a feed additive has been investigated to improve the reproductive and productive performance of different livestock species. However, their effects on the antioxidative defense systems of female reproductive structures are scarcely documented. The results obtained in the present study demonstrate that dietary supplementation of by-pass LO enhances the antioxidative system in luteal and uterine tissues on maternal recognition of the pregnancy period (on Days 14 and 16 after mating) in ewes.

**Abstract:**

This study analyzed the effects of dietary supplementation with by-pass linseed oil (LO; rich in α-linolenic acid) on maternal antioxidant systems at Days 14 and 16 of pregnancy in Sarda ewes. This trial used sixteen dry ewes. Eight ewes (CT group) were fed with a control diet without LO, and eight ewes (LO group) were fed with a diet supplemented with LO (10.8 g of α-linolenic acid/ewe/day). Both diets had similar crude protein and energy levels. The experiment included 10 days of an adaptation period and 31 days of a supplementation period. This supplementation period was divided into Period −2 (from Day −15 to −8), Period −1 (from Day −7 to −1; before synchronized mating period/Day 0), Period +1 (from Day +1 to + 7 after mating), and Period +2 (from Day +8 to +15 after mating). Estrous synchronization was induced in all the ewes using an intravaginal sponge (45 mg fluorgestone acetate) for 14 days and equine chorionic gonadotropin (350 UI/ewe) at the end of the treatment. On Days 14 (CT, N = 4; LO, N = 4) and 16 (CT, N = 4; LO, N = 4) after mating, the ewes were slaughtered. Samples of plasma, uterine, and luteal tissues were collected. Thiols, total antioxidant activity (TEAC), superoxide dismutase (SOD) activity, and malondialdehyde (MDA) content were measured. On Day 16, thiol and TEAC in luteal tissues were higher in the LO group when compared with the control one (*p* < 0.05). Moreover, TEAC was higher for the LO group in uterine tissues on Days 14 and 16 (*p* < 0.05). SOD activity was higher in the LO group in luteal and uterine tissues on Day 14 and Day 16, respectively (*p* < 0.001). On Day 16, uterine MDA content was lower for the LO group (*p* < 0.001). No differences were found between groups at the plasmatic level. However, the by-pass LO supplementation enhanced the analyzed antioxidant parameters in luteal and uterine tissues. In conclusion, these results demonstrate that by-pass LO supplementation exerted a positive effect on antioxidative defenses on maternal structures during the embryo-maternal recognition period in ewes. Thus, this could contribute to improving the maternal environment during the embryo-maternal recognition period in mammals.

## 1. Introduction

Long-chain polyunsaturated fatty acids omega 3 (PUFAs-ω3) and 6 (PUFAs-ω6) are well known for their beneficial role in several mammalian species [1,2]. They are involved in many biological processes such as vision, growth, brain development and reproduction [3,4]. Moreover, PUFAs are considered essential fatty acids because they cannot be synthesized by body tissues in mammals [5]. Therefore, they must be introduced through the diet [6]. The most important PUFA-ω3 has been found to be alpha-linolenic acid (ALA), having positive effects on the reproductive system in dairy cattle [7]. Fresh forages contain ALA concentrations ranging from 0.6 to 3.2% dry matter [8]. Its amount depends on seasonal changes, plant maturity, and the ruminal biohydrogenation process [9]. In large and small ruminants, one of the most valuable sources of ALA used as feedstuff is flaxseed/linseed oil (LO) [7,10]. A protected LO by-pass form has been used as a feed additive to avoid ruminal degradation (biohydrogenation) and guarantee its absorption in the intestine [11]. Most n-3 PUFAs are derived from ALA, which is the substrate for the synthesis of longer-chain and more unsaturated fatty acids such as ω-3 PUFA. PUFAs-ω3 as well as PUFAs-ω6 are integral components of the membrane lipid bilayer. The dietary supplementation and intake are reflected in the proportion of different PUFAs in different tissues, including the reproductive tract [4,6]. Many studies have shown that dietary PUFAs-ω3 supplementation can influence follicular development, ovulation, size, the number and function of corpora lutea (CLs), the synthesis of prostaglandin E_2_ (PGE_2_), steroidogenesis, and the maternal recognition of pregnancy and parturition. In this sense, the dietary supplementation of linseed oil has been linked to a reduction in endometrial prostaglandin F_2_α (PGF_2_α) synthesis, preventing luteal regression and guaranteeing progesterone (P_4_) synthesis and embryo survival [4,6,12].

The positive impact of PUFAs-ω3 on the antioxidant system has been reported in several human clinical trials, and these fatty acids are considered enhancer factors in reactive oxygen species (ROS) control [13].

Reported studies about the effects of PUFAs on the oxidative status in ruminant species are still limited and scarce.

To evaluate oxidative status, several parameters are usually studied. Thiol groups (SH of proteins and GSH) are water-soluble antioxidant molecules that have the ability to counteract radical propagation reactions and neutralize the action of the hydroxyl radical. Trolox equivalent antioxidant capacity (TEAC) quantifies antioxidant activity by non-enzymatic radical scavenging, and superoxide dismutase (SOD) is the most important enzymatic antioxidant. Malondialdehyde (MDA) is instead a product of lipid peroxidation and so a sign of oxidative damage.

In this sense, Didara et al. [14] reported greater plasmatic levels of MDA in periparturient dairy cows under dietary supplementation of linseed oil (source of PUFAs-ω3). Li et al. [15] observed an increment in SOD activity at the testicular/local level in ram fed with flaxseed oil. However, there is not any information on the effects of PUFAs-ω3 on the oxidative status of the female reproductive tract. In particular, studies about the antioxidative effects of PUFAs-ω3 on reproductive tissues as well as at the systemic level during the early stage of pregnancy could add valuable information to design therapeutic/prophylactic protocols to prevent early embryo losses and improve fertility in mammals [16].

Thus, the aim of this study was to evaluate whether the dietary supplementation of by-pass LO could enhance the antioxidative defenses of the reproductive tract, including luteal and uterine tissues, and at the plasmatic level during the embryo-maternal recognition period in dairy ewes.

## 2. Materials and Methods

The study was carried out from October to December at “Bonassai” experimental farm belonging to AGRIS-Sardegna (the agricultural research agency of Sardinia, Italy; 40°40′24″ North and 8°21′59.4″ East; 13–16 °C; 70–80% humidity). The experimental procedures were performed according to the rules of the Ethic Committee for Animal Lab Management from the Italian Health Ministry (Authorization n° 757/2020-PR/Protocol-E8652.2).

### 2.1. Animals and Treatment

Sixteen adult dry Sarda ewes were used in the study as previously reported by Contreras-Solís et al. [17]. All ewes were housed in pens with covered shelter (5 m^2^/ewe), under a natural light regimen and having free access to fresh water. Eight ewes were fed with a diet supplemented with by-pass LO (SILA™; Verona, Italy; 18% ALA; LO; N = 8), while the control group (CT; N = 8) was fed with a control diet without LO. Both diets were designed to guarantee similar crude protein (iso-proteic) and energy (iso-energetic) levels. The experimental period was 41 days. This period involved 10 days of adaptation (with the aim to obtain regular intake) and 31 days of full administration of experimental diets.

The supplementation period was organized into four groups: Period −2 (from day −15 to −1), Period −1 (from day −7 to −1; before synchronized mating/Day 0), Period +1 (from day +1 to + 7 after mating), and Period +2 (from day +8 to +15 after mating).

At the onset of the experiment, both CT and LO groups were homogenous in terms of age (CT = 1.71 ± 0.12 and LO = 1.48 ± 0.18 years old; *p* = 0.958), body weight (BW; CT = 38.23 ± 1.39 and LO = 38.45 ± 1.19 kg; *p* = 0.904), and body condition score (BCS; CT = 2.41 ± 0.11 and LO = 2.50 ± 0.08, *p* = 0.493).

Estrous synchronization was induced in all ewes using an intravaginal sponge (20 mg fluorgestone acetate; FGA, Chronogest; MSD, Igoville, France) for 14 days and equine chorionic gonadotropin (350 UI/ewe; Folligon, MSD, Boxmeer, The Netherland) at the end of the treatment. Ewes from both groups were mated using fertile rams 48 h after pessary withdrawal.

Ewes from the CT and LO groups were slaughtered on Day 14 (CT; N = 4; LO; N = 4) and 16 (CT; N = 4; LO; N = 4) after mating, to obtain luteal and uterine tissues.

### 2.2. Measurement and Sampling

#### 2.2.1. Body Weight and Body Condition Score

BW was measured using an electronic balance while BCS was measured using the score reported by Russel et al. [18].

BW and BCS were measured and recorded before the onset of the trial and each period from Period −2 to Period +2 (during the experimental period). Every measure was carried out from 8:00 to 9:00 before starting the feeding.

#### 2.2.2. Feedstuff Composition and Intake Measurement

The CT and LO diets were formulated to offer a non-fat and fat-enriched supplement, respectively, as already reported in cattle [19] and dairy sheep [17]. The chemical composition of experimental diets was previously described [17]. The CT group was fed with a daily mix of pelletized concentrate (300 g) and barley (250 g), while the LO group was fed with a mix of pelletized concentrate (250 g), barley (120 g), and bypass LO (60 g). The components of the two diets from each experimental group were weighted, mixed, and offered individually between 8:00 to 10:00. The orts were measured and recorded with the aim to calculate the individual intake.

In addition, daily hay (1125 g of Lolium multiflorum) was offered per animal for both the CT and LO groups. Daily intake was evaluated measuring remaining food 24 h later for each group. Individual average intakes of hay were calculated via dividing the daily intake per group by the number of ewes assigned to each experimental group.

#### 2.2.3. Blood Sampling

Blood samples were obtained from ewes belonging to each experimental group on Day 14 and 16 after mating (Period +2; onset of peri-implantational period). Blood samples were collected using vacuum collection tube with lithium heparin (VACUTEST KIMA srl—Via dell’Industria 12—35020 Arzergrande—PD—Italy). Immediately after recovery, blood samples were cooled at 4 °C, centrifuged at 1500× *g* for 15 min, and blood plasma was recovered and stored at −20 °C. Finally, these samples were analyzed to determine levels of thiols, TEAC, SOD, MDA, and P4.

#### 2.2.4. Tissues Samples

Ewes were euthanized using Pentothal Sodium (200 mg/kg body weight, MSD, Milan, Italy), administered intravenously via introducing a catheter into the jugular vein. Confirmation of animal death was performed through assessing the absence of a heartbeat, respiration, and reflexes. Then, luteal and uterine tissues were collected, weighed, and transferred in tubes with polyphosphate buffer (PBS). Then, clean tissues were transferred in tubes with 0.8 mL of PBS and 0.1% Triton-X 100 per 100 mg of tissue. Homogenization was performed using an electric homogenizer at room temperature. Finally, homogenized samples were centrifuged at 500× *g* for 5 min at +4 °C. Then, the supernatant was collected and stored at −80 °C, until the determination of thiols, TEAC, SOD, and MDA.

### 2.3. Antioxidant Markers

#### 2.3.1. Thiols Assay

A total thiols assay was performed in the blood plasma (50 µL) and extracts of luteal and uterine tissues (25 µL), adding 1 mL of Ellman’s Reagent 5,5-dithio-bis-(2-nitrobenzoic acid) (DTNB; 0.25 mM, D8130 Merk Life Science S.r.l., Milano, Italy) dissolved in phosphate buffer (0.3 M, pH 7.2) with the aim to release a yellow chromophore (5-mercapto-2-nitrobenzoic acid, Merk Life Science S.r.l., Milano, Italy) [20]. This chromophore was measured at 412 nm in a spectrophotometer (Thermo Elecrom Corporation Genesys 10 UV, Madison, WI, USA). Total thiols were quantified using a molar extinction coefficient of 13,600/M/cm and were expressed in µmol/g of protein for blood plasma and µmol/g of tissue for luteal and uterine tissues.

#### 2.3.2. Trolox Equivalent Antioxidant Capacity (TEAC) Assay

TEAC was measured using the method described by Re et al. [21] and modified by Lewinska et al. [22]. In this sense, 19.5 mg of 2,20-azino-bis(3-ethylbenzthiazoline-6-sulphonic acid) (ABTS, A3219 Merk Life Science S.r.l., Milano, Italy) and 3.3 mg of potassium persulfate (216224 Sigma-Aldrich Chemie GmbH, Taufkirchen, Germany) were dissolved in 7 mL of phosphate buffer (0.1 mol/L, pH 7.4, P2194 Merk Life Science S.r.l., Milano, Italy). This solution was incubated in darkness for 12 h to complete the reaction. Afterwards, this solution was diluted (1:80) in phosphate buffer (0.1 mol/L, pH 7.4) and mixed thoroughly to obtain an absorbance between 0.9 and 1.0 AU at 734 nm. Diluted samples (1:50) were added to the ABTS solution. After, the absorbance was spectrophotometrically measured twice at 734 nm.

The values of TEAC were expressed as nmol/g of protein for blood plasma and as nmol/g wet weight of tissue for luteal and uterine tissues.

#### 2.3.3. Superoxide Dismutase (SOD) Assay

SOD activity, for blood plasma, luteal and uterine tissues, was measured using an enzymatic method, as a decrease in the XTT (3′-(1-[(phenylamino)-carbonyl]-3,4-tetrazolium)-bis(4-methoxy-6-nitro) benzenesulphonic acid hydrate) (X750930, LubioSciernce GmbH, Zurich, Switzerland) reduction by superoxide anion generated by xanthine oxidase (X4376, Merk Life Science S.r.l., Milano, Italy) [23]. The SOD activity was assessed as the competition between two reactions and was measured as a decrease inthe rate of XTT reduced. The reaction mixture contained 40.5 mM sodium phosphate buffer pH 7.8 (P2194, Merk Life Science S.r.l., Milano, Italy), 15 mM xanthine (Thermo Scientific, Milano, Italy); EDTA 12.5 mM (Merk Life Science S.r.l., Milano, Italy), XTT 30 mM (LubioSciernce GmbH, Zurich, Switzerland), and 20 μL of sample to obtain a final volume of 500 μL. After the addition of xanthine oxidase (0.15 mUI), absorbance changes were monitored each minute for 3 min at 470 nm. The values of SOD were calculated using a standard curve (0.065–0.8 U/mL) and normalized to the protein content. One enzyme unit (U) is defined as the amount of SOD able to transform 1.0 mmol/min of O_2_^−^.

#### 2.3.4. Malondialdehyde (MDA) Assay

The thiobarbituric acid-reactive substances (TBARS) assay was used to measure MDA concentrations [24] with some modifications [25,26]. First, 100 μL of sample were added to mixed solution of 100 μL of 33% glacial acetic acid (Merk Life Science S.r.l., Milano, Italy), 75 μL of 10% sodium do-decyl sulfate (436143, Merk Life Science S.r.l., Milano, Italy), 100 μL of 50 mM Tris-HCl pH 7.4 (108315, Merk Life Science S.r.l., Milano, Italy), and 250 μL of 0.75% thiobarbituric acid (T5500, Merk Life Science S.r.l., Milano, Italy). After incubation for 1 h at 100 °C, the mixture was cooled on ice. After 10 min, 200 μL of 33% acetic acid was added, and samples were centrifuged for 20 min at 7000× *g*. The supernatant absorbance was read using a Thermo Electron Corporation Genesys 10 UV spectrophotometer at 535 nm.

A standard curve was used to calculate MDA values expressed as nmol/g of protein for blood plasma and nmol/g of tissue for luteal and uterine tissues.

### 2.4. Protein Assay

The total protein content in blood plasma was measured according to the Lowry method [27]. Alkaline solution (2% sodium carbonate in 1 N NaOH, 2% sodium tartrate 1% copper sulfate in water) and Folin’s reagent (SIGMA Folin–Ciocalteu′s phenol reagent F9252) were added to the sample in a 10:1:2 proportion. After 30 min at room temperature and in the dark, a spectrophotometer reading was taken at 750 nm. The concentration, expressed in µg/mL, was calculated based on the standard curve of albumin (2.5–40 µg/mL).

### 2.5. Progesterone (P4) Measurement

Plasma P4 concentration was assayed using a commercial kit (DRG Instruments GmbH, Marburg, Germany), a solid-phase ELISA with a competitive binding method [28]. All kit reagents, controls, and stored samples to be analyzed were thawed and warmed to 25 °C at the beginning of the test. The sample values were calculated through a calibration curve ranging from 0 to 40 ng/mL. The analytical sensitivity was 0.045 ng/mL, and the intra-assay and inter-assay CV values were <10%.

### 2.6. Statistical Analyses

Statistical assumptions (normality and homoscedasticity tests) were tested before further data analyses. Non-normal and heteroscedasticity data were analyzed using non-parametric statistical analyses.

Total diet intake (expressed in terms of NE), BW, and BCS on different periods were analyzed using repeated measures ANOVA. Antioxidant marker data (thiols, MDA, TEAC, and SOD activity) and P4 were analyzed using two-way ANOVA procedures. The models involved group, time, and their interaction.

Correlation analyses were performed to determine the association between local structures (luteal and uterine tissues) and blood plasma for each biomarker and P4.

Results were expressed as mean ± S.E.M. Statistical differences were established at *p* < 0.05. Statistical analyses were performed using R Studio.

## 3. Results

BW and BCS were similar between groups from the onset of the experimental period to Period +2. However, the time/period factor was significant for BW (*p* < 0.001, Table 1).

NE intake was similar between the groups (CT = 0.81 ± 0.01 and LO = 0.81 ± 0.01 Mcal/ewe/day; *p* = 0.244), as well as hay intake (CT = 0.66 ± 0.01 and LO = 0.66 ± 0.01 Mcal/ewe/day; *p* = 0.876). In consequence, total diet intake (in terms of NE) was similar between the groups in different periods (from Periods −2 to +2, Table 1).

Differences were found among the analyzed biomarkers linked to antioxidant activity in different tissues. Specifically, plasmatic levels of total thiols increased from Day 14 to 16 after mating for both the CT (*p* < 0.05) and LO groups (*p* < 0.01; Figure 1). However, on both Day 14 and Day 16, there were no statistically significant differences between the two groups in plasma (Figure 1). On Day 16, thiol levels in luteal tissue were higher in the LO group than in the CT group (*p* < 0.05; Figure 1), while on Day 14, there were no statistical differences between the groups. Regarding uterine tissues, no statistically significant differences were found in the two groups for both days.

Figure 2 shows the effect of treatment and time and their interaction on TEAC.

TEAC levels in luteal and uterine tissues increased (*p* < 0.01) from Day 14 to Day 16 in both experimental groups. Noteworthily, this increase was more marked in the luteal tissue from the LO group when compared to the CT group (*p* < 0.05). Moreover, in luteal tissue, on Day 14, there were no statistically significant differences between the two groups, while on Day 16, TEAC levels were higher in the LO group than in the CT group (*p* < 0.05). In uterine tissues, the LO group showed higher TEAC levels on both days when compared to the controls (*p* < 0.05). Regarding TEAC in blood plasma, there was no difference between days or even between groups.

SOD activity (Figure 3) in luteal and uterine tissues decreased significantly from Day 14 to Day 16 in both groups (*p* < 0.01 for CT and *p* < 0.001 for LO). Also, SOD activity was higher in luteal tissue on Day 14 (*p* < 0.01) for the LO group when compared with the CT group, while it was higher in uterine tissue (*p* < 0.01) on Day 16 for the LO group. Regarding SOD in blood plasma, there was no difference between days or even between groups.

MDA concentrations (Figure 4) in the luteal tissue decreased from Day 14 to Day 16 in both groups (*p* < 0.01 for CT and *p* < 0.05 for LO), while, in the same tissue, there was no difference between groups for the two days. MDA concentrations in uterine tissues increased in the CT group from Day 14 to Day 16 (*p* < 0.05). In this tissue, on Day 14, there was no difference between the groups, while on Day 16, MDA increased in the CT group and reached higher levels than in the LO group (*p* < 0.001). Plasmatic levels of MDA did not show differences between groups and days.

P4 concentration (Figure 5) in the plasma did not show any difference between the groups for the two days and between days.

Regarding the correlation analyzed for Day 14 (Table 2), in luteal tissue, thiols were positively correlated with TEAC (*p* < 0.01) and negatively with SOD (*p* < 0.01); also, TEAC was negatively correlated with SOD (*p* < 0.05) in the same luteal tissue. In blood plasma, only SOD was negatively correlated with MDA (*p* < 0.05). The other parameters had no correlation within the same tissue nor between the other analyzed matrices.

There was a significant positive correlation for Day 16 (Table 3), in luteal tissue, between thiol concentrations and TEAC (*p* < 0.05), while there were no correlations between thiols and the other parameters. Furthermore, thiols in luteal tissue were positively correlated with the SOD of uterine tissue (*p* < 0.05) and negatively with the TEAC of blood plasma (*p* < 0.05), while SOD in luteal tissue was negatively correlated with SOD plasma (*p* < 0.05) and P4 on Day 16 (*p* < 0.05). TEAC in luteal tissue was positively correlated with TEAC and SOD (*p* < 0.05 and *p* < 0,01) and negatively with the MDA of uterine tissue (*p* < 0.01).

In blood plasma, on Day 16 (Table 3), only thiols and SOD were significantly positively correlated (*p* < 0.05), while the other parameters were not significantly correlated; plasma SOD was negatively correlated with MDA in uterine tissue (*p* < 0.05). The other plasma parameters are not correlated among them.

Regarding uterine tissue on Day 16 (Table 3), TEAC and SOD were significantly positively correlated (*p* < 0.05), while TEAC and MDA were negatively correlated (*p* < 0.01); also, SOD was negatively correlated with MDA (*p* < 0.01) in the same tissue. The other uterine parameters were not significantly correlated within the same tissue nor between the other matrices analyzed.

## 4. Discussion

The main objective of the present study was to assess the effect of dietary supplementation of by-pass LO (rich in ALA/PUFAS-ω3) on oxidative parameters in female reproductive structures (luteal and uterine tissues), as well as in blood plasma during the period of maternal recognition of pregnancy in ewes.

In this trial, both CT and LO diets were designed to guarantee an isocaloric balance between experimental groups. Daily intake data indicate that these groups incorporated the same amount of energy as previously reported by Petit et al. [19,29] in cattle supplemented and non-supplemented with PUFAs. This lack of difference in energy intake is reinforced by the fact that BW and BCS were similar between groups during the experimental period.

Dietary supplementation with PUFAs-ω3 has been found to decrease oxidative stress (OS)-related mitochondrial dysfunction and endothelial cell apoptosis through the increased activity of endogenous antioxidant enzymes [30]. Furthermore, PUFAs-ω3 supplementation is able to restore imbalanced endogenous antioxidant systems, leading to a substantial antioxidant response [30].

When antioxidative defenses decrease, ROS increase and their eliminations diminish, determining a condition of OS [31]. Oxygen-free radical species are considered unstable and highly reactive molecules. They reach stability through acquiring electrons from lipids, proteins, and carbohydrates, yielding damage and diseases [32]. Lipid damage is often evaluated through measuring the lipid peroxidation product MDA. It is well known that ROS activity is involved in inflammatory and tissue injury processes as well as physiologic events such as ovulation and luteolysis [16,33,34].

PUFAs-ω3 are considered the most susceptible substrates of ROS during OS [35]. However, these fatty acids are also deemed as promoters of antioxidant defenses for ROS control [13].

Thiols (via glutathione and thioredoxin system) and SOD are involved in a variety of redox-regulated events mainly linked to enzymatic antioxidant activity [36], while TEAC is linked to non-enzymatic antioxidant capacity [25]. In our study, higher luteal SOD activity (on Day 14), as well as higher thiol and TEAC levels (on Day 16), were observed in ewes fed with by-pass LO when compared to the control group. This result suggests that the administration of by-pass LO, rich in the antioxidant PUFAs-ω3, promotes an increase in tissue antioxidant defenses. This result is confirmed via the correlations of the analyzed parameters. In fact, on Day 14 and 16, a positive and high correlation was observed between TEAC and thiols in luteal tissue and on Day 16 between TEAC and SOD in uterine tissues.

Major changes in antioxidant enzymes in the sheep luteal tissue during early pregnancy were reported by Al-Gubory et al. [37]. They suggested that the observed increase in antioxidant enzymes in this tissue is essential for protection from luteolysis, which could be caused by ROS continuously generated during luteal cells’ activity to produce steroids, like P4. In the present study, we observed an increase in SOD activity in treated ewes on Day 14 compared to controls. On Day 16, when SOD levels decreased in luteal tissue in both groups, the other antioxidant non-enzymatic parameters analyzed (TEAC and thiols) showed a significant increase in treated ewes when compared to controls. This increase could compensate for the decrease in antioxidant enzymes, thus ensuring the maintenance of a high level of antioxidant defenses in the luteal tissue. Taken together, these results may suggest that the administration of by-pass LO in the diet contributes to maintaining luteal activity and cellular integrity. The presence of ROS has indeed a negative effect on cellular function and could also affect P4 synthesis and luteal lifespan in mammals [38,39]. In the present study, we did not find any difference in P4 levels between the two groups. This result may be linked to the need to study luteal function for a longer period following pregnancy establishment. Further studies are thus needed to elucidate this issue.

Regarding uterine tissue, on Day 16, an MDA decrease was observed in the LO group when compared to the control group, suggesting a protective effect on membrane lipids, while TEAC increased in the LO group on the same day. Moreover, a negative correlation was observed between TEAC and MDA in uterine tissue. Our results highlight the capacity of by-pass LO (rich in PUFAS-ω3) to maintain controlled ROS levels also in uterine tissue [40], due to high levels of antioxidants. Ramos et al. [41] suggest that a defective uterine environment is associated with the inappropriate secretion of ovarian steroids and/or antioxidant enzyme activities, which contributes to early pregnancy failure. In this sense, the use of ω-3 supplements could support early pregnancy.

In blood plasma, we found negative correlations between SOD and MDA on Day 14, while positive correlations were observed between thiols and SOD on Day 16. Antioxidant biomarkers assessed at a plasmatic level were not affected by the dietary supplementation.

In luteal and uterine tissues, higher variations than in plasma were observed for the studied parameters. This is probably related to the differences between the analyzed matrices; in fact, the biomarker assay in tissues was performed on a homogenate matrix that included cells, whereas in blood, the assay was performed in plasma without including blood cells. Many papers [25,38,42] report comparisons of oxidative parameters in tissues and plasma, such as our work, but in light of our results, the effects of the dietary supplementation probably would have been more evident in blood cells than in plasma. More than 16 days after mating should be necessary to observe the mentioned positive effect of the antioxidant dietary supplementation in blood plasma. Alternatively, the use of by-pass LO in the present study could have an indirect effect on antioxidative defenses in the reproductive tissues. In fact, ROS may damage the luteal cell membrane and also affect progesterone production through interrupting trans-mitochondrial cholesterol transport [16,43]. However, PUFAs-ω3 promote progesterone synthesis (Gulliver et al., 2012) [6], which is associated with antioxidant defenses in reproductive tissues as referred by Al-Gubory et al. [39] and Chainy and Sahoo [44]. Thus, it could explain why the antioxidative effect of by-pass LO is more evident in tissues than at the plasmatic level. Therefore, additional studies should be conducted to elucidate differences among antioxidant biomarkers at systemic as well as at local levels, but it is clear that our results can be considered the first approach to establish a link between dietary supplementation of PUFAS-ω3 and ROS control on reproductive structures in sheep.

## 5. Conclusions

The results of the present study demonstrated that the dietary supplementation of by-pass LO exerted an antioxidative effect on reproductive tissues, increasing antioxidative defenses in local reproductive structures (luteal and uterine tissues) in early pregnancy. This suggests that increased antioxidant defenses guarantee protection from OS during the peri-implantation period and fertility in mammal species and could imply a possible helpful effect on the maternal environment regarding embryo-maternal recognition and embryo development. Further studies for a longer period during pregnancy could be useful to better elucidate the direct and/or indirect effect of PUFAS-ω3 at the local level.

## Figures and Tables

**Figure 1 animals-13-02565-f001:**
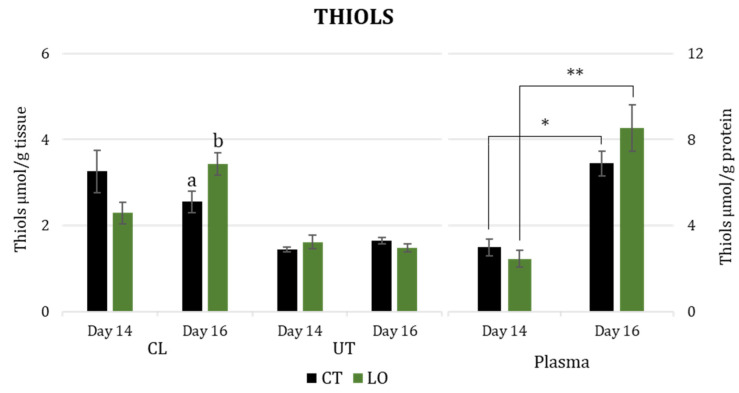
Thiol concentration (µmol/g of tissue) in luteal (CL) and uterine tissues (UT) and blood plasma (µmol/g of protein), at Days 14 and 16 after mating in dairy Sarda ewes non-supplemented (CT) and supplemented with by-pass linseed oil (LO) [Day 14 (CT; N = 4; LO; N = 4) and 16 (CT; N = 4; LO; N = 4)]. *, ** indicate differences between days at *p* < 0.05 and *p* < 0.01, respectively. a, b letters indicate differences between groups at *p* < 0.05.

**Figure 2 animals-13-02565-f002:**
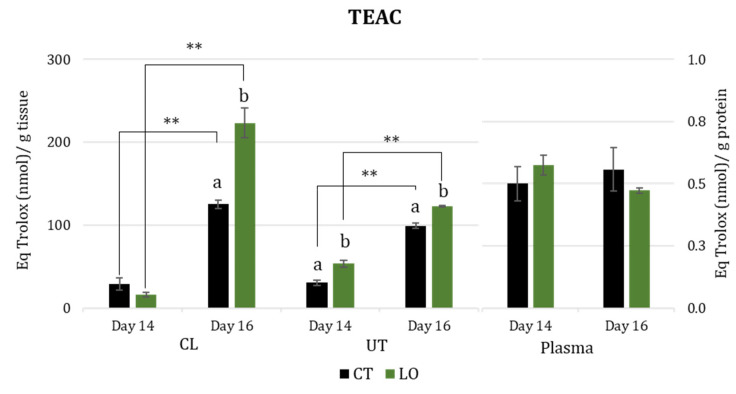
Trolox equivalent antioxidant capacity (TEAC) in luteal (CL) and uterine tissues (UT) (nmol/g of tissue) and blood plasma (nmol/g of protein), at Day 14 and 16 after mating in dairy Sarda ewes non-supplemented (CT) and supplemented with by-pass linseed oil (LO) [Day 14 (CT; N = 4; LO; N = 4) and 16 (CT; N = 4; LO; N = 4)]. ** indicate differences between days at *p* < 0.01. a, b letters indicate differences between groups at *p* < 0.05.

**Figure 3 animals-13-02565-f003:**
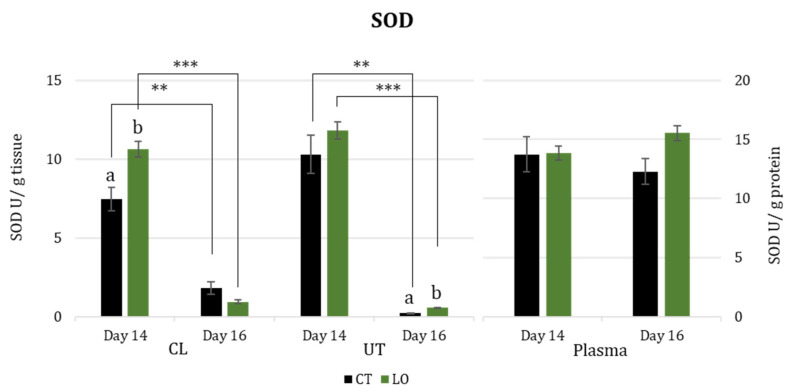
Superoxide dismutase activity (SOD;) in luteal (CL) and uterine tissues (UT) (U/g of tissue), and blood plasma (U/mg of protein), at Days 14 and 16 after mating in dairy Sarda ewes non-supplemented (CT) and supplemented with by-pass linseed oil (LO) [Day 14 (CT; N = 4; LO; N = 4) and Day 16 (CT; N = 4; LO; N = 4)]. **, *** indicate differences between days at *p* < 0.01 and *p* < 0.001, respectively. a, b letters indicate differences between groups at *p* < 0.001.

**Figure 4 animals-13-02565-f004:**
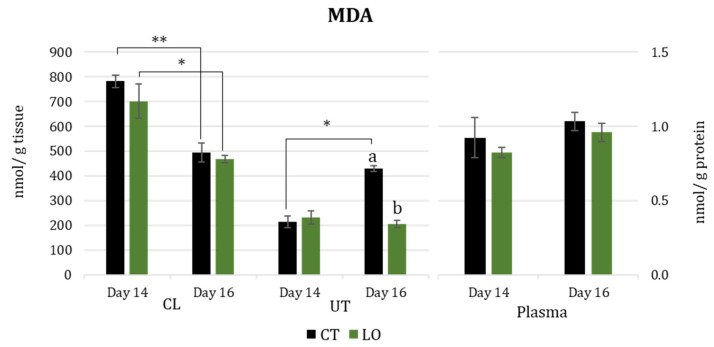
Malondialdehyde concentration (MDA) in luteal (CL) and uterine tissues (UT) (nmol/g of tissue) and blood plasma (nmol/g of protein), at Days 14 and 16 after mating in dairy Sarda ewes non-supplemented (CT) and supplemented with by-pass linseed oil (LO) [Day 14 (CT; N = 4; LO; N = 4) and Day 16 (CT; N = 4; LO; N = 4)]. *, ** indicate differences between days at *p* < 0.05 and *p* < 0.01, respectively. a, b letters indicate differences between groups at *p* < 0.001.

**Figure 5 animals-13-02565-f005:**
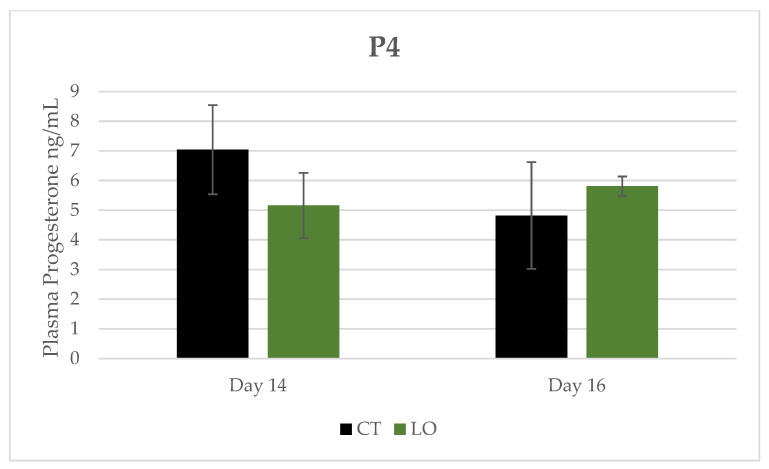
Progesterone (P4) concentration in blood plasma (mg/dL), at Days 14 and 16 after mating in dairy Sarda ewes non-supplemented (CT) and supplemented with by-pass linseed oil (LO) [Day 14 (CT; N = 4; LO; N = 4) and Day 16 (CT; N = 4; LO; N = 4)].

**Table 1 animals-13-02565-t001:** Body weight (BW), body condition score (BSC), and total net energy (NE; Mcal) intake/ewe/day during the experimental period in ewes non-supplemented (CT) and supplemented with by-pass linseed oil (LO) [Day 14 (CT; N = 4; LO; N = 4) and Day 16 (CT; N = 4; LO; N = 4)].

				*p* Value	
Variables	Group	Mean ± S.E.M.	Group	Time	Group × Time
BW (Kg)	CT	38.82 ± 0.64	n.s.	***	n.s.
	LO	39.08 ± 0.55			
BCS ^a^	CT	2.48 ± 0.03	n.s.	n.s.	n.s.
	LO	2.55 ± 0.03			
Total NE Intake (Mcal/ewe/day)	CT	1.47 ± 0.01	n.s.	n.s.	n.s.
	LO	1.47 ± 0.01			

^a^ score 1 to 5; n.s. = not significant; *** = *p* < 0.001.

**Table 2 animals-13-02565-t002:** Pearson correlation coefficients or Spearman rho correlation coefficients (*) between blood plasma progesterone concentration and thiols, TEAC, SOD, and MDA in luteal tissue (CL), uterine tissue (UT), and blood plasma on Day 14 (CT; N = 4; LO; N = 4).

	Thiols-CL	Thiols-UT	Thiols-PLASMA	TEAC-CL	TEAC-UT	TEAC-PLASMA	SOD-CL	SOD-UT	SOD-PLASMA	MDA-CL	MDA-UT	MDA-PLASMA
Thiols-UT	−0.452 (0.261)											
Thiols-PLASMA	−0.119 * (0.779)	−0.048 * (0.911)										
TEAC-CL	**0.938 (0.001)**	−0.486 (0.222)	0.119 * (0.779)									
TEAC-UT	−0.532 (0.175)	0.476 (0.233)	−0.357 * (0.385)	−0.416 (0.305)								
TEAC-PLASMA	0.144 (0.734)	0.562 (0.147)	−0.452 * (0.260)	0.026 (0.951)	0.448 (0.266)							
SOD-CL	**−0.869 (0.005)**	0.489 (0.219)	−0.333 * (0.420)	**−0.820 (0.013)**	0.657 (0.077)	0.039 (0.927)						
SOD-UT	−0.280 (0.502)	−0.258 (0.537)	−0.238 * (0.570)	−0.23 (0.584)	0.063 (0.883)	−0.323 (0.436)	0.556 (0.152)					
SOD-PLASMA	0.489 (0.219)	−0.011 (0.980)	−0.548 * (0.160)	0.509 (0.198)	−0.093 (0.827)	0.024 (0.956)	−0.240 (0.567)	0.116 (0.784)				
MDA-CL	0.177 (0.675)	−0.531 (0.176)	0.548 * (0.160)	0.274 (0.511)	−0.135 (0.75)	−0.394 (0.334)	−0.367 (0.372)	−0.325 (0.433)	−0.454 (0.259)			
MDA-UT	−0.225 (0.591)	0.552 (0.156)	0.286 * (0.493)	−0.268 (0.52)	0.414 (0.308)	0.359 (0.382)	0.204 (0.628)	−0.525 (0.182)	−0.363 (0.376)	0.331 (0.423)		
MDA-PLASMA	−0.512 (0.195)	−0.166 (0.694)	0.429 * (0.289)	−0.484 (0.225)	−0.268 (0.522)	−0.522 (0.185)	0.208 (0.621)	0.121 (0.775)	**−0.816 (0.013)**	0.390 (0.340)	0.020 (0.962)	
P4-DAY 14	−0.199 (0.636)	−0.196 (0.642)	0.381 * (0.352)	−0.338 (0.414)	−0.675 (0.067)	−0.513 (0.193)	−0.036 (0.932)	0.208 (0.621)	−0.214 (0.611)	−0.185 (0.660)	−0.451 (0.262)	0.605 (0.112)

Corresponding *p*-values are shown within parentheses. Correlations in the same tissue are highlighted with bold font.

**Table 3 animals-13-02565-t003:** Pearson correlation coefficients between blood plasma progesterone concentration and thiols, TEAC, SOD, and MDA in luteal tissue (CL), uterine tissue (UT), and blood plasma on Day 16 (CT; N = 4; LO; N = 4).

	Thiols-CL	Thiols-UT	Thiols-PLASMA	TEAC-CL	TEAC-UT	TEAC-PLASMA	SOD-CL	SOD-UT	SOD-PLASMA	MDA-CL	MDA-UT	MDA-PLASMA
Thiols-UT	−0.452 (0.261)											
Thiols-PLASMA	0.413 (0.309)	0.350 (0.396)										
TEAC-CL	**0.796 (0.018)**	−0.344 (0.405)	0.474 (0.235)									
TEAC-UT	0.498 (0.210)	−0.414 (0.308)	0.487 (0.220)	0.816 (0.014)								
TEAC-PLASMA	−0.716 (0.046)	0.408 (0.315)	−0.066 (0.877)	−0.449 (0.265)	−0.132 (0.756)							
SOD-CL	−0.543 (0.164)	0.022 (0.959)	−0.676 (0.066)	−0.617 (0.103)	−0.466 (0.244)	0.555 (0.153)						
SOD-UT	0.745 (0.034)	−0.575 (0.136)	0.379 (0.354)	0.877 (0.004)	**0.797 (0.018)**	−0.599 (0.117)	−0.692 (0.057)					
SOD-PLASMA	0.495 (0.212)	−0.018 (0.967)	**0.752 (0.031)**	0.698 (0.054)	0.605 (0.112)	−0.113 (0.789)	−0.776 (0.024)	0.676 (0.065)				
MDA-CL	0.183 (0.665)	−0.448 (0.266)	−0.586 (0.127)	−0.232 (0.580)	−0.463 (0.248)	−0.672 (0.068)	0.107 (0.801)	0.022 (0.959)	−0.487 (0.220)			
MDA-UT	−0.694 (0.056)	0.492 (0.215)	−0.546 (0.162)	−0.874 (0.005)	**−0.910 (0.002)**	0.334 (0.418)	0.610 (0.108)	**−0.935 (0.001)**	−0.769 (0.026)	0.268 (0.520)		
MDA-PLASMA	−0.172 (0.683)	0.570 (0.140)	0.005 (0.991)	−0.065 (0.879)	−0.159 (0.708)	0.357 (0.385)	0.503 (0.204)	−0.441 (0.274)	−0.336 (0.416)	−0.329 (0.426)	0.332 (0.421)	
P4-DAY 16	−0.466 (0.351)	0.270 (0.605)	0.650 (0.162)	0.066 (0.901)	0.237 (0.651)	0.165 (0.754)	−0.879 (0.021)	0.176 (0.739)	0.631 (0.179)	−0.628 (0.181)	−0.188 (0.721)	−0.766 (0.076)

Corresponding *p*-values are shown within parentheses. Correlations in the same tissue are highlighted with bold font. Correlations between different tissues are highlighted with underlined font.

## Data Availability

The data presented in this study are available on request from the corresponding author. The data are not publicly available due to a temporary lack of a publicly accessible repository.

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
