# Peer review of "The Effect of By-Pass Linseed Oil Supplementation on the Maternal Antioxidant System during the Embryo-Maternal Recognition Period in Ewes"

_animals, 2023, doi:10.3390/ani13162565_

Round 1
Reviewer 1 Report
Manuscript is interesting and rise important insights regarding the possible effect of PUFA on maternal anti-oxidant system, during embryo-maternal recognition period in ewes.
Yet, it suffers from some major problems; i beleive that most of them can easily be solved by revising the manuscript.
Currently the biggest problem in this manuscript is the partial and not entirely correct description and as of so the interpretation of the results.
Once, these issues will be solved. The manuscript is recommended for publication.
Introduction:
· Well written
Material and methods:
· In general, section should be re-edited to ensure proper scientific writing and clarity regarding: procedures and reagents, which not always present (intravaginal sponges, equine chorionic gonadotropin etc'.).
· Section 2.1: please add details regarding the ewes' husbandry (water, light regime, temperature, humidity and density).
· Section 2.2.4: please add details regarding the animals' euthanasia, prior to the tissues samples collection.
· Section 2.4: please add details regarding the protein assay.
Results:
· All findings presented in the results, should be addressed, even if no change was observed between treatments (for example: uterine samples and control group in the CL samples at day 14 in the thiols analysis and again between groups in the plasma sample , plasma samples in the TEAC analysis and in the MDA analysis.
Please revise the result section accordingly.
· Differences between groups should be presented in the text, even if they are not significant.
(For example: CL samples at day 14 between groups in the thiols analysis)
Please revise the result section accordingly.
· In the absence of correlation, I don't understand what is the additive value of presetting the correlation graphs? Moreover, when presenting correlation graphs, for each graph you should present: r, p and n values. It seems like the 2 sampled days were combined together in the correlation test – it may be better to perform the correlation tests for each day separately (for the tested markers, the behavior between days is not the same).
Discussion and conclusions:
·The discussion is very short and basically consists of 2 parts: first major part in which introduction is rewritten and second smaller part which discusses why the findings between the plasma and the tissues do not correlate.
This is a very shallow and insufficient discussion, that must be revised. The discussion, should refer to the entire findings presented in the results section (post this section revision). Currently, very important questions such as: what is the source for differences observed between 14 and 16 day in many of the markers? Why different markers behave differently?, remains unanswered.
Finally, the conclusions should be revised accordingly.
1. Extra spaces along the text: lines 38,230,335 - please revise.
2. Figure 2 label is superimposed on figure 2 legend - please revise.
3. lines 408-411 are duplicates of lines 343-346 - please delete them.
Author Response
Reviewer1 reply.
Dear Reviewer
Thank you very much for your efforts in the revision of our manuscript. Your comments and advice substantially improved the quality of the manuscript. All added and correct parts are in red in the resubmitted text.
Manuscript is interesting and rise important insights regarding the possible effect of PUFA on maternal antioxidant system, during embryo-maternal recognition period in ewes.
Yet, it suffers from some major problems; I believe that most of them can easily be solved by revising the manuscript.
Currently the biggest problem in this manuscript is the partial and not entirely correct description and as of so the interpretation of the results.
Once, these issues will be solved. The manuscript is recommended for publication.
Introduction:
- Well written
Material and methods:
Q1:· In general, section should be re-edited to ensure proper scientific writing and clarity regarding procedures and reagents, which not always present (intravaginal sponges, equine chorionic gonadotropin etc'.).
R1: The section has been re-edited as requested. Additional information about reagents used for oestrus synchronization was added in 2.1. section.
Q2: Section 2.1: please add details regarding the ewes' husbandry (water, light regime, temperature, humidity and density).
R2: The details regarding the ewes' husbandry have been added. Temperature and humidity data were included in the first paragraph of the “Material and Method” section. Also, housing/husbandry information has been described in 2.1. section.
Q3: Section 2.2.4: please add details regarding the animals' euthanasia, prior to the tissues samples collection.
R3: The details regarding the animals' euthanasia, prior to the tissue samples collection have been added.
Q4: Section 2.4: please add details regarding the protein assay.
R4: The details regarding the protein assay have been added.
Results:
Q5: All findings presented in the results, should be addressed, even if no change was observed between treatments (for example: uterine samples and control group in the CL samples at day 14 in the thiols analysis and again between groups in the plasma sample, plasma samples in the TEAC analysis and in the MDA analysis.
R5: All results have been now considered as requested.
Please revise the result section accordingly.
Q6: Differences between groups should be presented in the text, even if they are not significant.
(For example: CL samples at day 14 between groups in the thiols analysis)
R6: Differences between groups have been presented as requested, even if they are not significant.
Please revise the result section accordingly.
Q7: In the absence of correlation, I don't understand what is the additive value of presetting the correlation graphs? Moreover, when presenting correlation graphs, for each graph you should present: r, p and n values. It seems like the 2 sampled days were combined together in the correlation test – it may be better to perform the correlation tests for each day separately (for the tested markers, the behavior between days is not the same).
R7: Figures have been replaced with correlation tables 2 and 3 for days 14 and 16 respectively.
Discussion and conclusions:
Q8: The discussion is very short and basically consists of 2 parts: first major part in which the introduction is rewritten and second smaller part which discusses why the findings between the plasma and the tissues do not correlate.
This is a very shallow and insufficient discussion, that must be revised. The discussion, should refer to the entire findings presented in the results section (post this section revision). Currently, very important questions such as: what is the source for differences observed between 14 and 16 day in many of the markers? Why different markers behave differently?, remains unanswered.
R8: The “Discussion” section has been modified as you suggested.
Q9: Finally, the conclusions should be revised accordingly.
R9: The “Conclusion” section has been revised.

Reviewer 2 Report
The experiments were well conducted and the results properly analyzed. The paper is well written and easy to read and understand.
General comments
As mentioned by the authors, the beneficial effects of linseed oil are mainly related to its high content in ALA. But ALA acts mainly as precursor of other omega-3 fatty acids, like DHA, that are well-known to play important functions in reproduction. More information on the mechanisms by which the omega-3 fatty acids derived from ALA are beneficial from ovulation up to implantation could be added in the introduction.
The experimental design is not fully clear: 16 ewes entered the experiment, half were slaughtered at D14 and the remaining half at D16 post-oestrus to collect samples. Concerning the blood sampling were all the ewes collected at D14? While only the remaining 8 could be collected at D16. Please mention it more clearly as it can influence the results of the statistical analysis. The number of data per time point and treatment should be mentioned in the table and in the legends of the figures.
Only one data per sample and per analysis was used for the statistical analysis? Why not performing more than one analysis per kind of sample to evaluate the variability? For example, more than one corpus luteum is expected per ewe. Only one per ewe was used for the analysis ?
Concerning the (anti)oxidant analysis, each type of analysis carried out presents potential interpretation biases. A more critical discussion of the choice of the parameters and of the techniques themselves would be beneficial for the readers… and for a more nuanced interpretation of the results.
It seems that the authors split the data obtained during this experiment to publish them in two different papers ? This needs to be clarified. A correlation between the progesterone levels measured in the same animals (l467, ref16 manuscript in preparation) and the (anti)oxidant parameters would have added to the interest of the paper. The number and size of the corpora lutea as well as the presence and size of elongated embryos in the uterus deserve also to be evaluated and correlated with the other data.
L449: one ] is missing
Author Response
Reviewer2 reply.
Dear Reviewer
Thank you very much for your efforts in the revision of our manuscript. Your comments and advice substantially improved the quality of the manuscript. All added and correct parts are in red in the resubmitted text.
The experiments were well conducted and the results properly analyzed. The paper is well written and easy to read and understand.
General comments
Q1: As mentioned by the authors, the beneficial effects of linseed oil are mainly related to its high content in ALA. But ALA acts mainly as precursor of other omega-3 fatty acids, like DHA, that are well-known to play important functions in reproduction. More information on the mechanisms by which the omega-3 fatty acids derived from ALA are beneficial from ovulation up to implantation could be added in the introduction.
R1 Additional information about mechanisms by which the omega-3 fatty acids derived from ALA are beneficial from ovulation up to implantation has been added in the introduction.
Q2: The experimental design is not fully clear: 16 ewes entered the experiment, half were slaughtered at D14 and the remaining half at D16 post-oestrus to collect samples. Concerning the blood sampling were all the ewes collected at D14? While only the remaining 8 could be collected at D16. Please mention it more clearly as it can influence the results of the statistical analysis. The number of data per time point and treatment should be mentioned in the table and in the legends of the figures.
R2: Four ewes from each group were slaughtered on day 14. On day 16 other four ewes from each group were also slaughtered. It was clarified at the end of subparagraph 2.1. and at the start of subparagraph 2.2.3. Also, the information has been added to the table and in the legends of the figures.
Q3: Only one data per sample and per analysis was used for the statistical analysis? Why not performing more than one analysis per kind of sample to evaluate the variability? For example, more than one corpus luteum is expected per ewe. Only one per ewe was used for the analysis?
R3: One homogenised sample (per tissue and plasma) was analysed per ewe/day because other samples from these tissues were used for other analyses non-linked to this manuscript.
Q4: Concerning the (anti)oxidant analysis, each type of analysis carried out presents potential interpretation biases. A more critical discussion of the choice of the parameters and of the techniques themselves would be beneficial for the readers… and for a more nuanced interpretation of the results.
R4: The choice of the studied parameters was related to whether the treatment was able to affect their levels in plasma or tissues. Thiols groups (SH of proteins and GSH) are water-soluble antioxidant molecules that have the ability to counteract radical propagation reactions and neutralize the action of the hydroxyl radical. Trolox equivalent antioxidant capacity (TEAC) quantifies the antioxidant activity by non-enzymatic radical scavenging, and superoxide dismutase (SOD) is the most important enzymatic antioxidant. Malondialdehyde
(MDA) is instead a product of lipid peroxidation and so a sign of oxidative damage. These details have been added to the text.
Q5: It seems that the authors split the data obtained during this experiment to publish them in two different papers? This needs to be clarified. A correlation between the progesterone levels measured in the same animals (l467, ref16 manuscript in preparation) and the (anti)oxidant parameters would have added to the interest of the paper. The number and size of the corpora lutea as well as the presence and size of elongated embryos in the uterus deserve also to be evaluated and correlated with the other data.
R5: The plasma progesterone assay method was added in “Materials and Methods”; the assay results (for the two groups and for the two days) were reported in Figure 5 and in the “Results” section. Also, the correlations with the parameters studied have been added in the “Results” section and discussed in the “Discussion” section. As regards your additional suggestions, we believe that they are actually out of the scope of this work, and they are currently in use for drafting another manuscript where further experiments were carried out.

Round 2
Reviewer 1 Report
Material and methods:
· In general, section should be re-edited to ensure proper scientific writing and clarity regarding: procedures and reagents, which not always present – this issue was only partially fixed (mostly relevant to the reagents and kits manufacture data, used in the antioxidant assays).
· Lines 158-159 are not clear: " Plasma was removed and 157 stored at −20 °C until assayed, the levels of Thiols, Trolox equivalent antioxidant capacity 158 (TEAC), Superoxide dismutase (SOD), Malondialdehyde (MDA) and Progesterone (P4)." - Please revise.
Results:
· All findings presented in the results, should be addressed, even if no change was observed between treatments.
o Plasma MDA observations are not addressed - Please revise
· Tables 2&3, presents high amount of data – please highlight the ones, you want to put emphasis on (according to the text in lines 372-392.
Discussion and conclusions:
· Lines 444-446: "The observed decrease in luteal MDA in treated ewes compared to the controls observed on day 16 may suggest that a protective effect on membrane lipids is triggered" – this statement is not supported by the observations; no difference was found between groups regarding MDA at day 16 in the CL tissue –please revise.
· Lines 454-456: "On Day 16, when SOD levels decrease at tissue level in both groups, treated ewes showed a significant increase compared to controls in the other antioxidant non-enzymatic parameters analyzed in this study" – which tissues, do you refer to? Which other non enzymatic parameters, do you refer to? Please revise, so the statement would be accurate and clear.
· Lines 475-477: "Antioxidant biomarkers assessed at a plasmatic level were not affected by the dietary supplementation, except for thiols that increased on Day 16. when compared with their levels on Day 14" – please revise the sentence; the word "except" is not suitable, since there was no difference between groups for the thiols plasma levels at day 16.
· Lines 484-485: the sentence is not clear. Please revise.
English and editing:
Text must go through proofreading! Editing and grammars issues continue in the revised manuscript:
For example:
1. Line 56: the words having positive are connected.
2. Line 58: 2 commas without any word between them.
3. Lines 57, 60,182,209,211 and so on: extra spaces.
4. Line 259: the word "instead" is not suitable – please revise.
5. Line 263: the word group should be replaced with groups.
6. Some of paragraphs (especially in the results section, contain sentences in different tenses (past and present) – this should be revised.
Author Response
Reviewer reply.
Dear Reviewer
Thank you very much for your revision of our manuscript. All added and correct parts are in red in the resubmitted text. In track change, English editing has been reported. The "Author Contributions" section in the text, now complies with the "Authorship Change Form" sent to the journal.
Material and methods:
Q1. In general, section should be re-edited to ensure proper scientific writing and clarity regarding: procedures and reagents, which not always present – this issue was only partially fixed (mostly relevant to the reagents and kits manufacture data, used in the antioxidant assays).
R1: The section has been re-edited as requested.
Q2. Lines 158-159 are not clear: " Plasma was removed and 157 stored at −20 °C until assayed, the levels of Thiols, Trolox equivalent antioxidant capacity 158 (TEAC), Superoxide dismutase (SOD), Malondialdehyde (MDA) and Progesterone (P4)." - Please revise.
R2: The sentence has been revised.
Results:
Q3. All findings presented in the results, should be addressed, even if no change was observed between treatments.
• Plasma MDA observations are not addressed - Please revise
R3: The Plasma MDA observations have been added to the results section.
Q4. Tables 2&3, presents high amount of data – please highlight the ones, you want to put emphasis on (according to the text in lines 372-392).
R4: Data from Tables 2 and 3 were highlighted using bold font according to the text.
Discussion and conclusions:
Q5. Lines 444-446: "The observed decrease in luteal MDA in treated ewes compared to the controls observed on day 16 may suggest that a protective effect on membrane lipids is triggered" – this statement is not supported by the observations; no difference was found between groups regarding MDA at day 16 in the CL tissue –please revise.
R5: The sentence has been removed.
Q6. Lines 454-456: "On Day 16, when SOD levels decrease at tissue level in both groups, treated ewes showed a significant increase compared to controls in the other antioxidant non-enzymatic parameters analyzed in this study" – which tissues, do you refer to? Which other non enzymatic parameters, do you refer to? Please revise, so the statement would be accurate and clear.
R6: The sentence has been modified to improve its comprehension.
Q7. Lines 475-477: "Antioxidant biomarkers assessed at a plasmatic level were not affected by the dietary supplementation, except for thiols that increased on Day 16. when compared with their levels on Day 14" – please revise the sentence; the word "except" is not suitable, since there was no difference between groups for the thiols plasma levels at day 16.
R7. The sentence has been revised.
Q8. Lines 484-485: the sentence is not clear. Please revise.
R8. The sentence has been revised.
Q9. Comments on the Quality of English Language
English and editing:
Text must go through proofreading! Editing and grammars issues continue in the revised manuscript:
R9: The text has been proofread and editing and grammar issues have been revised.
For example:
1. Line 56: the words having positive are connected.
R. The words have been separated.
2. Line 58: 2 commas without any word between them.
R. One comma has been eliminated.
3. Lines 57, 60,182,209,211 and so on: extra spaces.
R. These lines have been arranged.
4. Line 259: the word "instead" is not suitable – please revise.
R. It was replaced by “However”.
5. Line 263: the word group should be replaced with groups.
R. It was corrected.
6. Some of paragraphs (especially in the results section, contain sentences in different tenses (past and present) – this should be revised.
R. All paragraphs have been revised.
